# Currently Applied Molecular Assays for Identifying *ESR1* Mutations in Patients with Advanced Breast Cancer

**DOI:** 10.3390/ijms21228807

**Published:** 2020-11-20

**Authors:** Nuri Lee, Min-Jeong Park, Wonkeun Song, Kibum Jeon, Seri Jeong

**Affiliations:** 1Department of Laboratory Medicine, Kangnam Sacred Heart Hospital, Hallym University College of Medicine, Seoul 07440, Korea; nurilee822@hallym.or.kr (N.L.); mjpark@hallym.or.kr (M.-J.P.); swonkeun@hallym.or.kr (W.S.); 2Department of Laboratory Medicine, Hangang Sacred Heart Hospital, Hallym University College of Medicine, Seoul 07440, Korea; pourmythe45@hallym.or.kr

**Keywords:** estrogen receptor, *ESR1*, breast cancer, next-generation sequencing, droplet digital polymerase chain reaction

## Abstract

Approximately 70% of breast cancers, the leading cause of cancer-related mortality worldwide, are positive for the estrogen receptor (ER). Treatment of patients with luminal subtypes is mainly based on endocrine therapy. However, ER positivity is reduced and *ESR1* mutations play an important role in resistance to endocrine therapy, leading to advanced breast cancer. Various methodologies for the detection of *ESR1* mutations have been developed, and the most commonly used method is next-generation sequencing (NGS)-based assays (50.0%) followed by droplet digital PCR (ddPCR) (45.5%). Regarding the sample type, tissue (50.0%) was more frequently used than plasma (27.3%). However, plasma (46.2%) became the most used method in 2016–2019, in contrast to 2012–2015 (22.2%). In 2016–2019, ddPCR (61.5%), rather than NGS (30.8%), became a more popular method than it was in 2012–2015. The easy accessibility, non-invasiveness, and demonstrated usefulness with high sensitivity of ddPCR using plasma have changed the trends. When using these assays, there should be a comprehensive understanding of the principles, advantages, vulnerability, and precautions for interpretation. In the future, advanced NGS platforms and modified ddPCR will benefit patients by facilitating treatment decisions efficiently based on information regarding *ESR1* mutations.

## 1. Introduction

### 1.1. Epidemiology of Breast Cancer

A total of 2,179,457 new breast cancer cases and 655,690 cancer-related deaths were estimated worldwide in 2020 according to the Global Cancer Observatory compiled and disseminated by the International Agency for Research on Cancer [1]. Among women, breast cancer is the most frequently diagnosed cancer and the leading cause of cancer-related deaths in major countries. Moreover, the incidence rates of breast cancer exceed those of other malignancies in both transitioned and transitioning nations [2,3]. The age-standardized incidence rates are highest in Australia/New Zealand; western, northern, and southern Europe; northern America; and southern and eastern Africa [4]. These international distributions are associated with cultural and/or environmental changes due to industrialization. Changes in nutrition, anthropometry, age at menstruation, hormone intake, lactation, and reproductive patterns, including fewer children and later age at first birth, contribute to the epidemiology of breast cancer [5].

### 1.2. Diagnosis and Molecular Heterogeneity of Breast Cancer

With respect to diagnosis and treatment, breast cancer is one of the most heterogeneous and complex diseases. Several subtypes with distinct biological characteristics result in differences in response patterns to diverse therapeutic regimens and clinical outcomes. Traditional classification systems have been based on tumor size, lymph node involvement, histological grade, age, and estrogen receptor (ER), progesterone receptor (PR), and human epidermal growth factor receptor 2 (HER2) status [6]. Approximately 75% of breast cancers are positive for ER and/or PR [6]. ER-positive tumors have responsive genes encoding typical proteins of luminal epithelial cells. Distinct intrinsic gene sets have been identified and combined into two main luminal-like subclasses corresponding to luminal A and luminal B to characterize the most common luminal subtype [7,8,9,10]. The luminal A subtype is defined as ER-positive and/or PR-positive, HER2-negative tumors with a low Ki-67 index [11]. Treatment of patients with luminal A breast cancer mainly involves endocrine therapy and has a good prognosis, with relapse and metastasis rates that are significantly lower than those of the other subtypes [12,13]. The luminal B subtype is defined as ER-positive, HER2-negative, with a high Ki-67 index or as ER-positive, HER-2-positive [14]. Luminal B cancers have a higher recurrence rate and lower survival rates than the luminal A subtype because they are relatively insensitive to endocrine therapy [15]. According to studies investigating molecular profiles [16,17], there are numerous subtype-associated gene mutations, including the enrichment of specific mutations in *GATA3*, *PIK3CA*, and *MAP3K1* in luminal-type breast cancers. The biological findings of the main breast cancer subtypes caused by different subsets of genetic and epigenetic alterations support the clinical observations of plasticity and heterogeneity within these major biological subtypes of breast cancer. Generally, these luminal subtypes presenting ERα encoded by the *ESR1* gene have a more favorable prognosis than those without. However, ER positivity was reduced 5–6 years after diagnosis, implicating an important role of resistance to endocrine therapy [18].

### 1.3. Endocrine Therapy and Resistance

Endocrine therapy has been recommended for patients with the luminal subtype of breast cancer [19]. Aromatase inhibitors, selective estrogen receptor modulators, and selective estrogen receptor degraders are major endocrine strategies for the treatment of ER-positive advanced breast cancers [20]. Aromatase inhibitors block the conversion of androgens to estrogens, resulting in lower levels of circulating estrogen and the resulting decreased stimulation of endocrine receptors. Selective estrogen receptor modulators such as tamoxifen bind both intracellular ERs and co-repressor proteins. It exhibits a partial ERα agonistic feature with net antagonistic activities on breast cancer tissue. Selective ER degraders, such as fulvestrant affect the stability of the ER and downregulate the receptor protein [21]. Although these medications reduce relapse rates when administered prophylactically after surgery and chemotherapy, resistance to endocrine therapy leading to the development of fatal metastatic cancer commonly occurs and has become a major clinical problem [22]. When approximately 15% of patients with ER-positive breast cancer relapse lose ER expression, targeting the ER is likely to be ineffective. The remaining 85% of patients might progress to metastatic status because of the inevitable acquired resistance. Resistance to endocrine therapy is an important driver of mortality due to metastasis to distant organs.

#### 1.3.1. Mechanisms of Resistance to Endocrine Therapy

The mechanisms of resistance to endocrine therapy are complex and have been investigated previously [23]. Aberrantly activated growth factor receptor tyrosine kinases, such as epidermal growth factor receptor (EGFR) and HER2 (ERBB2) have increased ER transcriptional activity in a hormone-independent manner [24]. ER-positive cases that express ERBB2 amplification have reduced sensitivity to ER-targeted endocrine therapies, leading to poor outcomes [25]. However, clinical trials using EGFR inhibitors have shown modest or negative results [26]. Dysregulation of the cyclin D-CDK4/6-Rb axis promoting resistance to endocrine therapy is frequently observed in the luminal B subtype [16,27]. CDK4/6 inhibitors, namely palbociclib and ribociclib, are approved for use in combination with endocrine therapy for treating advanced ER-positive cancers [28]. In particular, *ESR1* was more frequently mutated in metastatic breast cancer than in early breast cancer and has been identified both as a driver and as a metastatic gene [29]. Alterations in *ESR1*, such as amplifications, point mutations, and chromosomal rearrangement events, have been identified as well-known mechanisms in driving endocrine therapy resistance and metastatic disease progression.

#### 1.3.2. *ESR1* Mutation

The *ESR1* gene located at 6q25.1–q25.2 encodes an ER and a ligand-activated transcription factor consisting of several domains involved in hormone binding [30]. Related pathways include the estrogen signaling pathway and signaling by G protein-coupled receptors, the large family of cell surface receptors [31]. Various *ESR1* mutations are implicated in hormone resistance and anti-estrogen therapies, such as tamoxifen, aromatase inhibitors, and fulvestrant, in patients with ER-positive breast cancer [32,33,34]. Resistance to endocrine therapy derived from *ESR1* can be classified into acquired and de novo patterns. *ESR1* expression changes over time; thus, negative results at a point of disease evolution can become detectable at another time [20]. According to studies investigating *ESR1* mutations, there is persistent activation of ER regardless of its ligand. A shift in helix 12 of *ESR1*, leading to the similarity to the estrogen-bound activated state of ER, was suggested as a mechanism for this ligand-independent ER activity. Coactivators may be able to bind and activate ER because of a change in ER configuration conferring resistance to endocrine therapy [35]. This mutational mechanism found in approximately 40% of patients with metastatic breast cancer who were pretreated with aromatase inhibitors [36,37]. Various *ESR1* alterations, amplification, genomic rearrangement, and point mutations contribute to the therapeutic resistance and metastasis of ER-positive breast cancer.

##### *ESR1* Amplification

*ESR1* amplification is identified in approximately 30% of patients with ER-positive breast cancer depending on detection techniques and scoring systems [38,39,40,41,42]. A positive association between *ESR1* amplification and ER protein expression demonstrated in several studies suggests that amplification results in the increased production of oncogenic ER proteins [41,42,43]. In terms of the clinical significance of *ESR1* amplification, the link between the presence of *ESR1* amplifications in breast malignancy and resistance to endocrine therapy leading to metastasis is not clear and needs further investigation. *ESR1* amplification in a subset of ER-positive breast cancers was correlated with tamoxifen resistance and poor prognosis of patients [18,44]. In contrast, *ESR1* amplification was suggested as an indicator of longer disease-free survival and elevated sensitivity to endocrine therapy in contradicting studies [42,45]. Further studies are necessary to fully understand the clinical implications of *ESR1* amplifications owing to these controversial results.

##### *ESR1* Rearrangements and Fusion

The genomic rearrangements of *ESR1* have also been investigated. Diverse *ESR1* gene fusion transcripts have been identified in luminal breast cancer cases [46,47]. According to a previous study, RNA-sequencing data from primary TCGA breast samples demonstrated that 2.1% of all luminal B subtype samples harbored recurrently fused transcripts. The identified transcripts involved in the first two non-coding exons of *ESR1* fused to various C-terminal sequences from the coiled-coil domain containing 170 genes (*CCDC170*) (*ESR1*-e2 > *CCDC170*). These fusion transcripts generate truncated forms of CCDC170 proteins, which cannot complete chimeric ER fusion proteins. Therefore, exogenous expression of these truncated CCDC170 proteins in ER-positive breast cancer cells results in overgrowth and decreased sensitivity to tamoxifen [47]. This study presented a representative role for *ESR1*-e2 > *CCDC170* in endocrine therapy resistance. In terms of metastatic ER-positive breast cancer, *ESR1* fusions follow a similar fusion pattern harboring the first six exons of *ESR1* (*ESR1*-e6) fused to the C-terminus of various gene partners. Therefore, this pattern is considered important for endocrine therapy resistance based on *ESR1* fusion structural studies. However, detailed functional characterization and evidence supporting a causal role for *ESR1* fusions have been lacking. Moreover, the incidence of *ESR1* fusions is estimated at approximately 1%, and the exact value has not yet been established [48]. Additional studies that provide evidence for the causal role of *ESR1* fusions, as well as their significant diagnostic and clinical implications need to be performed. A recent study [49] showed a novel mechanism that *ESR1*-*CCDC170* bound to HER2/HER3/SRC and activated SRC/PI3K/AKT signaling. Therefore, treatment regimens combining endocrine agents with the HER2 inhibitor lapatinib and/or the SRC inhibitor dasatinib might be applied to patients with *ESR1*-*CCDC170* gene fusions. Furthermore, kinase fusions in breast cancer analyzed by Memorial Sloan Kettering-Integrated Mutation Profiling of Actionable Cancer Targets seemed to be enriched in hormone-resistant, metastatic carcinomas and mutually exclusive with *ESR1* mutations [50]. Based on these results, fusion testing as a molecular testing at progression after endocrine therapy was suggested in an effort to identify additional therapeutic options which may provide substantial clinical benefit.

##### *ESR1* Point Mutation

Among several mechanisms of *ESR1* mutation, the acquisition of activating point mutations, which cluster within the ligand-binding domain (LBD) of *ESR1,* is a well-known mechanism for acquired endocrine therapy resistance. Substitution of tyrosine at position 537 to serine (Y537S) in the LBD of *ESR1* conferring constitutive, ligand-independent activity of ER was first reported in experimental breast cancer models [51]. Regarding human tumors, Y537N was found in a metastatic specimen from a patient with breast cancer who experienced disease progression while on endocrine therapy [52], suggesting its ability to drive resistance to tamoxifen. The three most frequently identified *ESR1* point mutations were D538G, Y537N, and Y537S [20,34]. *ESR1* LBD point mutations mostly affect Y537 and D538 residues [40,48]. Samples from patients with ER-positive breast cancer treated with endocrine therapy rather than those from treatment-naïve patients revealed these mutations [48], supporting a role for *ESR1* mutations in acquired resistance to endocrine therapy and metastasis [53].

In vitro studies have been performed to characterize the functional, transcriptional, and pharmacological properties of *ESR1* LBD point mutations. Cell line models expressing *ESR1* mutants showed that these mutants contribute to hormone-independent proliferation resistant to endocrine treatment [34,40,54,55]. These mutations are in an apo-receptor conformation, which are constitutively active, even upon antagonist binding [34,56]. Changes in protein structure derived from these *ESR1* mutations resulted in reduced ligand affinity and ligand-independent activity. Although fulvestrant inhibited the growth of point mutation-containing cells in a dose-dependent manner, growth was not reversed to the levels of wild-type *ESR1*-expressing cell lines [40,54]. When bound to fulvestrant, the *ESR1* mutant also showed enhanced protein stability compared to the wild-type receptor. Moreover, these *ESR1* LBD mutations also recruited coactivators that further potentiated ER transcription [55,57]. In terms of metastatic biology, *ESR1* mutant cell lines, including Y537S and D538G, presented a substantial enrichment of metastasis-associated gene sets. The Y537S mutant showed remarkably potentiated tumor growth and metastasis in patients treated with tamoxifen or fulvestrant compared to the D538G mutant because of different cistromes and transcriptomes [53]. Regarding signaling pathways activated by *ESR1* mutants, interactions between ERs and receptor tyrosine kinases, including EGFR, HER2, and insulin-like growth factor receptor, activate downstream kinases. Particularly, co-localization and crosstalk between mutant ER and the insulin-like growth factor receptor pathway were revealed using ER immunoprecipitation and proximity ligation assays [58]. This upregulated insulin-like growth factor receptor pathway was demonstrated in *ESR1* mutant overexpression models [32,59]. These related pathways induce phosphorylation of multiple transcription factors, such as ERs and co-factors, leading to gene expression in a hormone-independent manner [60], suggesting a role for mutant ERs in promoting a metastatic phenotype [53,61]. With respect to the therapeutic strategy for preventing *ESR1* mutant-driven breast tumors, targeting these signaling pathways could be considered. According to a recent study [62], the emergence of circulating *ESR1* mutations was related to the risk of early progression during aromatase inhibitor treatment in patients with metastatic breast cancer, which suggested the potential role of *ESR1* mutations as a useful biomarker. In addition, the immunogenicity of epitope derived from the most common *ESR1* mutations including D538G and Y537S was suggested as novel targets for breast cancer immunotherapy [63].

### 1.4. Aims of This Review for Molecular Assays

Several molecular methods have been developed and evaluated for these *ESR1* mutations that are strongly involved in breast cancer development and progression [20,64]. Variations in different assays used in diverse studies affect their results and clinical utility. When adopting these assays, a comprehensive understanding of the principles, advantages, disadvantages, and precautions for data interpretation should be well-recognized. Articles focusing on laboratory methods for evaluating *ESR1* mutations have been rarely published. Therefore, our study reviews molecular assays for *ESR1* mutations, the main cause of advanced estrogen-positive breast cancer with resistance to endocrine therapy. Advances in techniques according to the epoch are also investigated based on published articles for gaining insight into molecular methodologies and for facilitating appropriate laboratory settings for evaluating *ESR1* mutations, which are useful for predicting outcomes and planning patient-tailored therapeutic strategies.

## 2. Sample Type

The sample types obtained from patients with breast cancer for detecting *ESR1* mutations using molecular assays are presented in Table 1. Twenty-two included studies were from three recently published systematic reviews and meta-analyses [20,64,65] reporting *ESR1* mutations in advanced breast cancers with resistance to endocrine therapy. The most frequently used sample type was tissue (50.0%), including formalin-fixed paraffin-embedded (FFPE) (66.7%) and fresh frozen (25.0%) tissues (Figure 1a). Archival FFPE tissues have been a powerful resource for NGS studies [66]. However, formalin fixation results in intra- and inter-molecular crosslinks. These crosslinks create difficulties in efficient DNA extraction and amplification. Acidic and non-buffered formalin also induces DNA fragmentation. Sequence artifacts, such as an increase in the C > T/G > A change at low to moderate allele frequency levels of <10%, have been observed [67,68]. Therefore, evidence-based best practices for FFPE DNA extraction provided by the National Cancer Institute [69] and European Committee for Standardization Technical Specifications on the pre-analytical phase of FFPE DNA have been published [70]. Tissue ischemia time and temperature, fixation time, storage conditions of the FFPE blocks and DNA extraction, and pre-normalization concentration of libraries should be carefully considered based on previous studies [71]. Among several kits for tissue DNA extraction, QIAamp DNA FFPE Tissue Kit and DNeasy Blood & Tissue Kit from Qiagen [70] are the most frequently used.

Plasma samples in K2/K3 ethylenediaminetetraacetic acid (EDTA), Streck Cell-free DNA blood collection, and heparinized tubes accounted for 27.3% of the sample type. The concurrent use of tissue and plasma specimens constituted 22.7% of the total sample types. Although heparinized tubes have been used in some studies, they are not recommended for polymerase chain reaction (PCR) because heparin is a well-known inhibitor, and residual heparin can influence PCR amplification [72]. According to a study comparing cell-free DNA (cfDNA) in different types of tubes, significantly higher cfDNA concentrations were detected in EDTA samples than in heparinized samples. Moreover, cfDNA concentrations were more stable over time within the EDTA matrix than heparin [73]. EDTA blood should be processed within 2–6 h at room temperature and should be kept frozen at −20 °C or −80 °C for longer than 24 h [74,75]. Streck Cell-free DNA blood collection tubes provided by Streck Inc. showed even more stable plasma cfDNA concentrations after storage for 2–14 days than EDTA tubes [76,77]. These tubes are strongly recommended, particularly for multicenter NGS studies because of their beneficial effect on plasma cfDNA [73]. Among studies using droplet digital PCR (ddPCR), the QIAamp circulating nucleic acid kit was most commonly used for DNA extraction.

The differences in the sample types used for detecting *ESR1* mutations according to previous publications are illustrated in Figure 2a. Plasma (46.2%) became the most commonly used method in 2016–2019, in contrast to 2012–2015 (22.2%). The easy accessibility, non-invasiveness, and usefulness of plasma samples with the increased use of ddPCR have influenced these trends.

## 3. Trends in Molecular Assays

The molecular assays used and main characteristics observed in the studies of *ESR1* mutations are presented in Table 1. The included studies were mainly based on three meta-analyses for *ESR1* mutations in patients with breast cancer with resistance to endocrine therapy [20,64,65]. The search strategy terms for these systematic review and meta-analyses were as follows: (‘estrogen receptor alpha’ OR ‘*ESR1*’) AND (‘breast neoplasms’ OR ‘breast cancer’) AND (‘neoplasm metastasis’ OR ‘metastasis’ OR ‘resistance’). Relevant articles with molecular assays for *ESR1* mutations were also manually checked. A total of 22 studies were included to reflect assays currently used in clinical laboratories [16,17,29,33,34,35,36,40,55,78,79,80,81,82,83,84,85,86,87,88,89,90]. The most used molecular assay was NGS (50.0%) followed by ddPCR (45.5%) (Figure 1b). Targeted NGS gene panels are designed for a specific disease or a group of diseases, with the ability to maximize coverage, sensitivity, and specificity of the genes of interest. Therefore, targeted NGS gene panels often have higher diagnostic yields than exome sequencing or genome sequencing. One study reporting *ESR1* mutations in bone metastases from breast cancer adopted NGS and ddPCR simultaneously [25]. NGS was performed for *ESR1* coding regions, and all detected mutations were confirmed using pyrosequencing. The possible *ESR1* amplification was also tested using ddPCR. Commercially available platforms for NGS were utilized (Illumina (San Diego, CA, USA) HiSeq 2000 or Hiseq 2500 (83.3%) or Thermo Fisher Scientific Personal Genome Machine (PGM) instrument) (16.7%)). For library preparation, hybridization capture with biotinylated RNA-based oligonucleotide probes from Agilent and barcoded sequence libraries provided by Illumina were frequently utilized. Paired-end sequencing was performed in most studies. For ddPCR, all included studies utilized platforms provided by Bio-Rad Laboratories except for two studies using the QuantStudio 3D Digital PCR System [91] from Thermo Fisher Scientific [35,78]. Generally, Sanger sequencing is used for confirming detected mutations in these molecular platforms. The most detected *ESR1* mutation was D538G, followed by Y537 residues. Studies from Eastern countries are necessary because most studies have been conducted in Western countries.

The changes in molecular assays for *ESR1* mutations in advanced breast cancer according to the epoch of examination are presented in Figure 2b. In 2016–2019, ddPCR (61.5%) was more frequently used than NGS (30.8%), and vice versa in 2012–2015. The advent of ddPCR focusing on *ESR1* mutations with high sensitivity could influence changes in user preferences.

**Table 1 ijms-21-08807-t001:** Molecular assays and main characteristics of the studies for advanced breast cancer with *ESR1* mutations.

First Author [Year]	Tumor Type (*n*)	Sample Type	Detection Method	Sequencing Equipment of Kit (Company)	Most Frequent ESR1	Study Country
**NGS based**						
Bartels et al., (2018), [78]	BC with bone metastases (231)	FFPE	NGS and ddPCR	Ion PGM Hi-Q Kit v2 using 318 v2 Chips and QuantStudio 3D Digital PCR System (Thermo Fisher Scientific, Germany)	D538G	Germany
Cancer Genome Atlas, (2012), [16]	Luminal BC (169)	Tissue	NGS and several methods	Illumina (Illumina, USA)	NA	Multi-national
Ellis et al., (2012), [17]	Luminal BC (46)	Snap-frozen tissue	NGS	Illumina (Illumina, USA)	NA	USA
Jeselsohn et al., (2014), [40]	Metastatic BC (76)	FFPE	NGS	HiSeq2000 (Illumina, USA)	D538G and Y537N	USA and Spain
Lefebvre et al., (2016), [29]	Metastatic BC (143)	Fresh frozen tumor biopsy	NGS	Illumina HiSeq2500, HiSeq4000, or NextSeq500 (Illumina, USA)	NA	France
Merenbakh-Lamin et al., (2013), [55]	Metastatic BC (13)	FFPE	NGS	Illumina HiSeq2000 (Illumina, USA)	D538G	Israel
Nik-Zainal et al., (2016) [83]	BC (560)	FFPE	NGS	Illumina GAIIx, Hiseq 2000 or Hiseq 2500 (Illumina, USA)	NA	Multi-national
Niu et al., (2015) [36]	Metastatic BC (222)	FFPE	NGS	Illumina HiSeq2000 platform (Illumina, USA)	Codon Y537	USA
Robinson et al., (2013) [33]	Metastatic BC (11)	Frozen needle biopsy	NGS	Illumina HiSeq2000 platform (Illumina, USA)	NA	USA
Toy et al., (2013) [34]	Advanced BC and Metastatic BC (36)	Fresh frozen tissue and FFPE	NGS	Illumina Hiseq 2000 (Illumina, USA)	D538G	USA
Toy et al., (2017) [87]	Metastatic BC (265)	FFPE	NGS	Illumina HiSeq 2500 (Illumina, USA)	D538G	USA
Yanagawa et al., (2017) [89]	Primary BC (16) and recurrent BC (46)	FFPE and plasma	NGS	Torrent PGM instrument (Thermo Fisher Scientific, USA)	D538G	Japan
**ddPCR based**						
Chandarlapaty et al., (2016) [79]	Metastatic BC (541) related to BOLERO-2 clinical trial	Plasma in EDTA	Single ddPCR	QX200 Droplet Digital PCR System (Bio-Rad Laboratories, USA)	D538G	USA
Chu et al., (2016) [80]	Metastatic BC (23)	Plasma in Streck BCT DNA tube or EDTA	ddPCR	QX200 Droplet Digital PCR System (Bio-Rad Laboratories, USA)	D538G	USA
Clatot et al., (2016) [81]	BC with progression (144)	Plasma in heparinized tube	Single ddPCR	QX200 Droplet Digital PCR System (Bio-Rad Laboratories, USA)	D538G	France
Gyanchandani et al., (2017) [90]	Relapsed or metastatic BC (16)	Plasma in Streck Cell-free DNA blood tubes	ddPCR	QX100 Droplet Digital PCR System (Bio-Rad Laboratories, USA)	D538G	USA
Fribbens et al., (2016) [82]	BC with relapse or progression (161) related to SoFEA and PALOMA-3 clinical trials	Plasma in EDTA	Multiplex and uniplex ddPCR	QX200 Droplet Digital PCR System (Bio-Rad Laboratories, USA)	D538G	USA
Schiavon et al., (2015] [84]	Advanced BC (171)	Plasma in EDTA or Streck Cell-Free DNA BCT tube, and FFPE	Multiplex ddPCR	QX200 Droplet Digital PCR System (Bio-Rad, USA), Ion AmpliSeq Breast Cancer Panel (Thermo Fisher Scientific, USA), and PI chip using the Ion PI OT2 200 Kit (Thermo Fisher Scientific, USA)	D538G	United Kingdom
Sefrioui et al., (2015) [35]	Metastatic BC (7)	Frozen pleural biopsy, FFPE for primary tumor sample, and plasma in heparinized tube	ddPCR	QuantStudio 3D Digital PCR System (Thermo Fisher Scientific, USA)	NA	France
Spoerke et al. (2016) [85]	Metastatic BC (153) related to FERGI clinical trial	Plasma and FFPE	ddPCR	QX200 Droplet Digital PCR System (Bio-Rad Laboratories, USA)	D538G	USA
Takeshita et al., (2017) [86]	Advanced BC (17) and Metastatic BC (69)	Plasma in EDTA	Single ddPCR	QX200 Droplet Digital PCR System (Bio-Rad Laboratories, USA)	Y537N	Japan
Wang et al., (2016) [88]	Primary or metastatic BC (29)	Frozen tissue and plasma in Streck tubes	ddPCR	QX100 Droplet Digital PCR System (Bio-Rad Laboratories, USA)	D538G	USA

BC, breast cancer; FFPE, formalin-fixed paraffin-embedded; NGS, next-generation sequencing; ddPCR, droplet digital polymerase chain reaction.

## 4. Molecular Assays

### 4.1. Next-Generation Sequencing

NGS has offered a quantum leap in the field of molecular assays. The NGS technique provides significant improvements in sequencing speed and throughput because of the automated streamlined workflow. The overall workflow consists of template preparation such as nucleic acid extraction, library preparation, sequencing, bioinformatics, and data interpretation. Several manufacturer-specific platforms, such as the Illumina HiSeq and Ion Torrent PGM systems have been developed for identifying *ESR1* mutations.

#### 4.1.1. Library Preparation

The preparation of an NGS library starts with the fragmentation of nucleic acids. Physical techniques such as acoustic shearing, sonication, and hydrodynamic shear or enzymatic methods, including digestion by deoxyribonuclease I and Fragmentase, have been used for fragmentation [92,93]. Sequence adaptors, which create known starts and ends, are connected to fragments for enrichment. Fragments are selected according to the desired library size. When DNA libraries contain shorter fragments of similar sizes, short-read sequencers adopted most commonly in the included studies yielded the best results. Illumina fragments (up to 1500 bases) are longer than Ion Torrent PGM fragments (up to 400 bases) [92,94]. The shorter fragments are feasible for targeted and exome sequencing rather than whole-genome sequencing, as most of the human exons are less than 200 base pairs in length [95]. The following enrichment step increased the amount of target DNA for sequencing. Capture hybridization-based sequencing and amplicon-based sequencing exist for targeted strategies [96,97]. The former method utilizes biotinylated oligonucleotide probes, and fragmented materials are hybridized physically to DNA fragments complementary to the targeted regions. The widely used SureSelect kit (Agilent) for library preparation in the included studies is based on this hybrid capture methodology [94]. For amplicon-based methods, synthetic oligonucleotides with a sequence complementary to the flanking region of the target DNA were utilized. The representative commercially available kit is Ion AmpliSeq from Thermo Fisher Scientific. The limitations of this amplicon-based preparation originate from PCR amplifications, including primer competition, duplicates, and non-uniform amplification of target regions because of the variation in GC content. Hybrid capture techniques have advantages concerning uniform coverage and depth over amplicon-based methods. However, hybrid capture techniques require more time and higher costs than amplicon strategies [96,97]. To overcome these limitations, barcoded sequence libraries using unique molecular identifiers ligated to DNA fragments were developed [98] for sequencing *ESR1* mutations using TruSeq of Illumina [34]. PCR duplicates were detected using non-unique fragments combined with unique molecular identifiers [98,99].

#### 4.1.2. Sequencing Platforms

##### Illumina

Illumina platforms are the most widely used in our included studies for detecting *ESR1* mutations. Illumina developed a bridge PCR strategy for clonal amplification and sequencing through a reversible terminator. Fragmented DNA is attached to fixed adapters on the solid surface of the flow cell. Bridge amplification forms double-strand bridges followed by denaturation to generate single-stranded templates. After repeating this process, several millions of dense clusters containing clonal fragments are generated in each channel of the flow cell. A reversible terminator, a single labeled complementary deoxynucleotide triphosphate, retains a cleavable fluorescent dye. When reversible terminators are added to the template during synthesis, fluorescent signals are captured and recorded during each cycle [100]. This methodology substantially reduces the homopolymer sequencing error because the addition of a single base at a time is required [100,101]. However, some fluorescent dyes may have a partial overlap between the emission spectra of the fluorophores during the process. Moreover, they may lose activity, limiting the base calling on this Illumina platform [102,103].

The most used Illumina sequencers in the included studies for *ESR1* mutation were HiSeq2000 and Hiseq 2500, launched in 2012 [104] (Table 1). Both platforms belong to second-generation sequencing technologies and utilize four-channel sequencing using a synthesis system in which each fluorescently labeled base is detected using individual images. Their common read lengths were 100 × 100 base pairs. The maximal output and run time were 600 GB and 11 d, respectively. The difference between HiSeq2000 and Hiseq 2500 is the fast run mode adopted in the Hiseq 2500 system. When the fast run mode is operated, the read length becomes 150 × 150 base pairs. Additionally, the run time can be reduced to 27 h in the fast run mode [105]. NextSeq 500 and HiSeq 4000 systems, launched in 2014 and 2015, respectively, were also utilized for detecting *ESR1* mutations [29]. NextSeq 500 utilizes two-channel sequencing using the synthesis system, which requires only two images to measure all four base calls. This new methodology decreases the sequencing cost and time by reducing the number of cycles and imaging capture time. HiSeq 4000 systems adopted billions of pre-formatted nanowell grids at fixed locations, leading to a much higher data output than previous platforms using normal flow cells [101]. Altogether, Illumina systems have been used most commonly in laboratory settings for detecting *ESR1* mutations because of their high throughput, accuracy, and relatively low cost.

##### Ion Torrent

Ion Torrent employs semiconductor sequencing utilizing hydrogen ion detection technology. The fragmented molecules are attached to the beads with specific adapter sequences. Amplification using emulsion PCR generates beads with clonally amplified target DNA [106]. Each bead is then loaded into the microwells on the semiconductor sensor array chip. When nucleotides are incorporated into growing strands, protons are released, causing pH changes, and the ion sensor on the complementary metal-oxide-semiconductor chip detects these signals [107,108].

The first Ion Torrent PGM sequencer, launched in 2010, was most frequently used in laboratories for detecting *ESR1* mutations (Table 1) [78,89]. The PGM has a read length of approximately 400 base pairs (200 × 200 base pairs for paired-end sequencing). Maximal output is up to 1 GB per run, and the run time is fast (2–7 h) [32,33]. The Ion Torrent platform covers long reads and is more direct, more rapid, and less expensive than Illumina, which relies on laser scanners. However, these sequencers yield relatively low output and are vulnerable to insertion/deletion (indel) errors associated with homopolymeric stretches and repeats [109]. The observed raw error rate of Ion Torrent PGM (1.71%) was higher than that of Illumina HiSeq 2000 (0.26%) [110]. Advanced models of second-generation sequencing platforms with higher throughput, easier preparation, and shorter run time have been developed and will be utilized in future studies for identifying *ESR1* mutations.

#### 4.1.3. Bioinformatics

The huge amount of NGS data generated using sequencing platforms requires computational and bioinformatics skills for management, analysis, and interpretation. Bioinformatics has been developed and improved based on algorithms, software applications, and increased computational capacities of hardware. Basic procedures for analyses are common; however, this study focuses on Illumina and Ion Torrent, the two major second-generation platforms, and each system has its own particularities and specificities.

First, the detection and analysis of raw data, base calling, and scoring base quality (Phred score) are required. Typical outputs from this step are the fastq file of Illumina and the unmapped binary alignment map file from Ion Torrent. Fastq files containing all the raw sequencing reads, file names, and quality values are essential for the first quality control step [111]. Several bioinformatics tools have been developed for evaluating the quality of raw data, such as the NGS QC toolkit [112], QC-Chain [113], and FastQC [111]. FastQC, one of the most popular tools, provides a report presenting well-structured and graphically illustrated information about quality. Additional trimming is conducted at the ends of each read to eliminate adapter sequences. Among some trimming tools for Illumina data, appropriate tools are selected according to the dataset, downstream analysis, and parameters used [114]. For Ion Torrent, sequencing and data management are processed in the Torrent Suite software. After trimming, the removal of remaining library adapter sequences from the ends of the reads, demultiplexing was performed. This step is required to prevent interfering with mapping and assembly owing to residual adaptor sequences in the reads [114].

The next step is read alignment against the reference human genome (typically hg19 or hg38), which is preferred in clinical settings rather than de novo assembly for mapping sequence reads [115]. Burrows–Wheeler Aligners and Bowtie are the most widely used alignment software programs for Illumina data, whereas the Torrent Mapping Alignment Program is recommended for Ion Torrent [116,117]. As output, a binary alignment/map (BAM) containing a text file format and a sequence alignment/map (SAM) format having binary versions can be obtained after analyses [118]. The Interactive Genome Viewer [119] is available for checking the alignments. Post-alignment processing for reducing the base call and alignment artifacts is necessary for improving the variant call accuracy and quality of the downstream process [120]. SAMtools [118], Genome Analysis Toolkit (GATK) [121], and Picard [94] are useful tools for post-alignment processing. The variant calling step is processed for identifying variants using tools such as SAMtools, GATK, and Freebayes for Illumina data [122] and Torrent Variant Caller for Ion Torrent. A variant calling format (VCF) file can be generated via these tools, mostly from the SAM/BAM format as the input [123]. The VCF file contains meta-information lines, a header line, and data lines presenting information about the chromosomal position, reference base, and identified alternative base. The last analysis starts with variant annotation, which inserts a further layer of information into all the variants identified in previous variant calling steps to predict their functional impacts. Several annotation tools based on functionality include SIFT [124], PolyPhen-2 [125], CADD [126], and Condel [127], which calculate the consequence scores for each variant based on the degree of conservation of amino acid residues, evolutionary conservation, sequence homology, protein structure, or statistical prediction derived from known mutations. For the clinical associations, ClinVar and HGMD, disease variants databases, can be utilized. After annotation, steps for making clinical sense and identifying disease-causing variants via some filtering strategies are required.

After variant annotation, steps for variant filtering, prioritization, and data visualization should be followed. A combination of several software programs is required to conduct these analytical steps. The total number of independent reads and the percentage of reads presenting the variant and the homopolymer length (especially for Ion Torrent) could be applied as quality parameters for decreasing variant call errors and the number of false-positive calls. The user should determine the threshold, such as 10 independent reads, based on the observed data for filtering sequencing bias or low coverage. Additionally, minor allele frequency [128] and population databases, such as the 1000 Genomes Project [129], Exome Aggregation Consortium [130], and Genome Aggregation Database can be powerful tools for filtering. The representative American College of Medical Genetics and Genomics and the European Society of Human Genetics guidelines provide standards and guidelines for the evaluation and interpretation of genomic variations from NGS [131].

#### 4.1.4. NGS strategies

##### Targeted Panel Sequencing

Targeted panels designed for specific genes can maximize coverage, sensitivity, and specificity for the included genes. Therefore, targeted panel sequencing provides a higher diagnostic yield than whole exome sequencing or whole-genome sequencing. The cost of targeted panels is usually lower than that of whole exome sequencing. Niu et al. [36] used targeted NGS platforms for 315 cancer-related genes and 28 genes commonly rearranged in cancer. *ESR1* mutations are one of the genomic alterations reported using this targeted panel.

##### Whole Exome Sequencing

Whole exome sequencing analyzes all exons, and clinical exome sequencing targets approximately 22,000 protein-coding genes. These sequencing strategies increase the chances of detecting pathogenic variants. Patients with negative results in targeted panel sequencing or complex phenotypes are usually indicated for exome sequencing [104]. Among the included studies, a study investigating the association between *ESR1* mutations and efficacy of ER antagonists [87] utilized a commercially available MSK-IMPACT assay, a targeted sequencing assay covering all protein-coding exons and introns of 410 key oncogenes. Other studies evaluating the mutational profile of metastatic breast cancers [29] and comprehensive molecular portraits of breast malignancies [16] adopted whole exome sequencing using Illumina platforms.

##### Whole Genome Sequencing

Whole genome sequencing is a comprehensive technique for analyzing the entire genome. It can detect variants missed using targeted panel sequencing or whole exome sequencing, providing increased diagnostic yield [104]. In terms of costs, a single test ranged from 555–5169 USD for exome sequencing and 1906–24,810 USD for whole genome sequencing based on a recently published meta-analysis on the cost-effectiveness of different sequencing platforms [132]. Several studies on *ESR1* mutations treated with endocrine therapy utilized whole exome sequencing using Illumina platforms [16,17,36].

### 4.2. Droplet Digital PCR

After second-generation PCR, such as real-time quantitative PCR quantifying the target molecules with standard curves, the revolutionary ddPCR as the third-generation PCR was developed and commercialized in 2011. This methodology allows absolute quantification via partitioning of the reaction [133]. Therefore, it is applied for the detection of *ESR1* mutations because of its high sensitivity and accuracy.

Briefly, molecules are separated randomly into numerous small-volume compartments, such as water-oil emulsion droplets (emulsion-based digital PCR) or a chip with microchannels (microfluidics-based digital PCR) automatically. The distribution of target sequences in the compartments can be calculated using Poisson statistics. Each compartment functions as an individual PCR micro-reactor. The amplified target sequences are identified using fluorescence, and the initial copy number and concentration of the target molecule can be obtained. This method provides a template to approximately 20,000 droplets before amplification, showing high sensitivity for finding mutations in DNA with a detection limitation of 0.001% [134]. The numerous micro-compartments contribute to the increase in the tolerance of the PCR system to inhibitors. Based on this high sensitivity compared to 1% of traditional PCR, it has been applied for the detection of circulating tumor DNA. Furthermore, ddPCR has been technologically modified into a commercial tool that is more operable and compatible. Its two-dimensional data are highly readable [135]. According to the representative scatter plot of ddPCR results of the Bio-Rad QX200 ddPCR system, which was mostly utilized in our studies for identifying *ESR1* mutations, the mutant allele appeared to be blue and the wild-type reference allele was shown in green using fluorescent probes. Double-positive droplets harboring both types of molecules are presented in orange, whereas double-negative droplets with no amplification are presented in gray. Moreover, ddPCR generates reliable results with small amount of samples because of the elimination of error from pre-amplification. Therefore, it enables the utilization of samples with degraded nucleic acids and samples that are difficult to obtain [136]. Furthermore, ddPCR provides high-repeatability results based on its independence of amplification efficiency. It defines the amplification results as positive and negative and quantifies the results with counts assisted using Poisson statistics. The drawbacks of ddPCR include a narrow dynamic range and a relatively higher cost than real-time quantitative PCR. However, the cost of ddPCR is lower than that of NGS technology.

Based on our included studies, ddPCR technologies are increasingly being used for the detection of *ESR1* mutations in advanced breast cancers owing to the enhanced sensitivity of ddPCR, which enables more accurate positive detection of *ESR1* mutations in plasma samples. The Bio-Rad QX200 and QX 100 ddPCR systems are the most widely utilized. In the BOLERO-2 clinical trial, allele-specific assays for *ESR1* D538G and Y537S mutations were designed and optimized for quantification using the QX200 ddPCR system. The extracted cfDNA was analyzed for the presence of *ESR1* mutations using single ddPCR assays [79]. Fribbens et al. [82] used both multiplex and single ddPCR assays to identify the seven most common *ESR1* mutations from the SoFEA and PALOMA-3 samples. If at least two *ESR1* mutant droplets are observed, a multiplex assay is determined to be positive. The results from the multiplex ddPCR format are further characterized using single ddPCR assays. Schiavon et al. [84] also developed multiplex ddPCR assays by varying the concentration of fluorescent probes for distinguishing mutations based on fluorescence intensity. A study of *ESR1* mutations in patients with ER-positive metastatic breast cancer treated with fulvestrant [85] designed ddPCR probe assays for 10 *ESR1* mutations in samples from the FERGI clinical trial. Wang et al. [88] assessed six *ESR1* mutations (K303R, S463P, Y537C, Y537N, Y537S, and D538G) using ddPCR (QX100 Droplet Digital PCR System) with a lower limit of detection of 0.05–0.16%. Studies on *ESR1* mutations in circulating plasma tumor DNA [80,86,90] and for evaluating the prognostic and predictive values of *ESR1* circulating mutations [81] employed Bio-Rad ddPCR systems. Two studies employed the QuantStudio 3D Digital PCR System from Thermo Fisher Scientific [35,78]. Bartels et al. [78] performed ddPCR using chip-based nanofluidics to assess possible *ESR1* gene amplification and for confirming NGS variant detection. The application of ddPCR to circulating tumor DNA for detecting *ESR1* mutations and monitoring treatment response or progression is still being investigated. Several clinical trials, such as BOLERO-2 [79], SoFEA, PALOMA-3 [82], and FERGI [85], also used ddPCR, demonstrating a need for the routine use of targeted sequencing of liquid biopsies.

### 4.3. Other Methods

#### 4.3.1. Sanger Sequencing

The first-generation platform, Sanger sequencing developed by Fred Sanger in 1977 [137] was the primary sequencing technology for the subsequent two and a half decades. It has long been considered the reference standard for nucleic acid sequencing. Sanger sequencing generates relatively long (500–1000 base pairs), high-quality DNA sequences [138]. Sanger sequencing and ER-specific exon PCR were utilized to identify an *ESR1* mutation, Y537N, from a metastatic ER-positive tumor biopsy in 1997 [52]. Recently, Sanger sequencing has been used to confirm mutations identified using NGS or ddPCR technology. DNA from a cell line with a D538G knock-in mutation and that from a liver biopsy with an *ESR1* mutation at Y537S showing high allele frequencies were confirmed using Sanger sequencing [88]. In another study, mutations found in recurrent breast tumors with more than 10% mutant allele frequency via NGS were confirmed using the Sanger method with a commercially available ABI PRISM 310 Genetic Analyzer (Applied Biosystems, Foster, CA, USA) [89]. Sanger sequencing has been utilized mainly for confirmation because it requires much more time and cost than NGS technology.

#### 4.3.2. Pyrosequencing

The advent of pyrosequencing technology by Roche 454 sequencing (Roche, Switzerland) in 2005 began the NGS revolution [139]. It was the first commercially available massive parallel sequencing platform that generates thousands to millions of short sequencing reads (140–200 bases for paired end reads) in a single machine run without cloning [105]. Pyrosequencing methodology captures pyrophosphate release and uses it as an indicator of specific base addition. After binding fragmented DNA to beads through ligated adaptors, emulsion PCR within an emulsion droplet results in fragment amplification [106]. The beads harboring multiple copies of the same DNA template are loaded into PicoTiterPlate wells. Each nucleotide is sequentially flowed into the wells. During DNA synthesis, a nucleotide is incorporated, and the pyrophosphate released is converted to ATP. Luciferase converts luciferin to oxyluciferin to generate light in the presence of ATP. The signal is then detected and captured using a coupled-charge device camera [139,140]. Sequencing accuracy relies on the reading of the light signals. A misread or missing signal related to homopolymer sequencing leads to base errors, insertions, or deletions. Roche 454 was replaced with emerging NGS platforms, such as the Ion Torrent (Thermo Fisher) and Illumina systems, in 2016, owing to its higher cost. Among our included studies, pyrosequencing was performed for sequencing *ESR1* exon 5. In addition, it was also used to confirm all mutations in *ESR1* identified using the Ion Torrent NGS platform [78].

#### 4.3.3. Real-Time PCR

The real-time PCR technique, a prior form of ddPCR, has been used in several genetic platforms for patients with breast cancer. This method requires the amplification of specific DNA targets using PCR and monitoring of the amplification reaction using fluorescence. It is routinely used for the analysis of gene expression and quantification [141]. Reverse transcription of RNA into cDNA is frequently coupled with this type of PCR. Multiplex PCR targeting various molecules based on real-time PCR is also performed using commercial products. In the oncology field, it is one of the methods for measuring minimal residual disease in patients with breast cancer [142]. Jeselsohn et al. [40] adopted this method for measuring mRNA levels after synthesizing cDNA using a reverse transcription kit. Although real-time PCR involves relatively shorter run and hands-on times and is more sensitive and provides more reliable results than traditional PCR, [143] one should consider the possible disadvantages, such as false-positive results owing to non-specific amplification, contamination, and false-negative results owing to low gene expression.

#### 4.3.4. Microarray

The microarray technique, which is a collection of microscopic DNA or RNA spots attached to a solid surface, has been used for a small number of breast tumor tissues. Microarrays are utilized as a high-throughput method for measuring the expression of a large number of genes, including the *ESR1* gene [144]. In breast cancer, it is used to quantify relative transcript abundance. In addition, MammaPrint, a genetic platform, is a microarray-based assay for determining prognosis in breast cancer using the expression levels of 70 genes to assess the risk of recurrence [145]. Among the included studies, Ellis et al. [17] validated all putative somatic mutations via targeted custom capture arrays. In another study, Agilent mRNA expression microarrays and Illumina Affymetrix 6.0 single-nucleotide polymorphism arrays containing more than 906,600 probes were adopted for the comprehensive molecular portraits of breast malignancies [16].

#### 4.3.5. Methylation

DNA methylation, a reversible epigenetic change, plays an important role in the carcinogenesis, progression, and prognosis of various malignancies. Genes with aberrant promoter methylation are known to be closely associated with breast cancer development and progression [146]. Methylation mediates the silencing of genes and drives speculation regarding the role of molecular crosstalk between genes or genetic pathways. According to a recently published meta-analysis [147], *ESR1* promoter methylation is related to a worse overall survival of patients with breast cancer. This meta-analysis included three studies on *ESR1* methylation. A study on DNA methylation profiles of breast cancer was performed using the MethyLight assay [148] for DNA methylation analysis. After sodium bisulfite-converted genomic DNA amplification, fluorescence was identified using a laser detector. The percentage of fully methylated molecules at a specific locus was calculated [149]. In the remaining two studies [150,151], methylation-specific PCRs were performed to analyze the methylation status of methylated genes in breast cancer tissues. Two sets of primers were designed for the targeted gene. One was specific for DNA methylation at the promoter region and the other was specific for unmethylated DNA. After amplification, each PCR product was identified. Additionally, Illumina Infinium DNA methylation arrays and the main NGS platform were used for the molecular profiles of breast malignancies [16].

## 5. Future of Molecular Assays

Advanced types of NGS techniques and modified forms of ddPCR exhibiting micro-sized instruments, ultra-sensitive and specific detectors, faster turnaround times, and lower costs will be developed in the future for detecting *ESR1* mutations more efficiently. Although second-generation NGS technologies have substantially impacted this field, short sequence reads resulting in sequence gaps, alignment issues related to repetitive regions or pseudogenes, and PCR artifacts remain to be addressed [94]. Third-generation sequencers based on single-molecule sequencing techniques have been developed to overcome these limitations [152,153]. The representative commercial platforms are single-molecule real-time Sequel systems from Pacific Biosciences (Menlo Park, CA, USA) and MinION from Oxford Nanopore Technologies (Oxford, UK) [154]. PacBio platforms (Pacific Biosciences) allow much longer sequencing reads (average, 10–15 kb), shorter sample preparation (4–6 h), shorter sequencing run time (within a day/run), and reduced GC bias and sequencing errors derived from PCR amplification. However, it has a relatively high error rate (10–15%) owing to indels caused by miscalls. The nanopore sequencing platform relies on electrical changes as each nucleotide passes through the nanopore [155]. Nanopore technology exhibits short turnaround time and no GC bias and is independent of amplification steps requiring fluorescence labeling and DNA polymerase. The most apparent drawback of this technology is the high sequencing error rate of 14%, which stems from errors from indels. Altogether, the third-generation platforms provide longer sequence reads, which can contribute to sequencing of extended repetitive elements and characterizing pathogenic structural variations in advanced breast tumors. The disadvantages of this technology might be complemented by combining second- and third-generation NGS methodologies [156].

The BEAMing method is a modified type of digital PCR that is based on beads, emulsion, amplification, and magnetics. Water droplets in an oil emulsion serve as reaction vessels and contain primers, a mixture of template, PCR reagents, and magnetic beads. Fluorescently labeled dideoxynucleotide terminators are utilized to differentiate droplets containing sequences of interest and are analyzed via flow cytometry [157]. BEAMing exhibited a highly sensitive detection rate of 0.02% mutant allele frequency and a specificity of 100%, with a small amount of required DNA input and a higher than 90% concordance rate between tissue and circulating tumor DNA from the breast [157]. Moreover, good agreement between BEAMing and ddPCR and applying BEAMing to a large-scale clinical trial could be evidence of sufficient reproducibility for clinical use [158,159]. Despite its high performance, the complicated workflow and high cost should be overcome for routine clinical work. Additionally, Lupini et al. [160] developed enhanced-ice-COLD-PCR with high sensitivity and demonstrated the utility of this approach with respect to simplification and improvement for detecting *ESR1* mutations. Furthermore, a commercially available Guardant360 assay based on an NGS assay utilizing circulating tumor DNA to identify genomic alterations among 54–70 cancer-related genes, including 18 copy number variants and six gene fusions, was also adopted for personalized cancer treatment for breast tumors [161].

Genetic profiling of putative genetic factors associated with resistance to endocrine therapy using molecular tools such as NGS and digital PCR might improve patient prognosis and increase accuracy in predicting responses to therapeutic interventions. Furthermore, rapid and accurate detection of *ESR1* alterations based on the advent of newly developed methods for differentiating subtypes and emerging variants resistant to currently used therapies will greatly benefit patients with advanced breast cancers. Analytical and clinical validation should be conducted before using new techniques in clinical laboratories.

## 6. Conclusions

In summary, applied molecular methodologies for identifying *ESR1* mutations, one of the major causes of resistance to endocrine therapy in patients with advanced breast cancer, were reviewed. The main methods used for detecting *ESR1* mutations in laboratories are NGS-based assays, including Illumina and Ion Torrent platforms. This is followed by ddPCR, including single and multiplex assays, accounting for approximately half of the molecular methods. Regarding sample types used for molecular assays, tissues such as FFPE and fresh frozen samples were once the major resources for detecting *ESR1* mutations. However, comparison of the sample types using a wide range of available data showed that plasma obtained from EDTA, Streck Cell-free DNA blood, and heparinized tubes have become the more appropriate specimens according to the changes in the adopted molecular assays. The predominant method for detecting *ESR1* mutations involves ddPCR in the recent epoch. Although NGS-based assays are still in use because of their wide dynamic range for detecting mutation alterations, ddPCR offers higher sensitivity, economic feasibility, and highly readable and repeatable results. Moreover, easier accessibility because of non-invasiveness and minimal sample volumes have influenced the preference for ddPCR. The advent of ddPCR facilitates routine investigations of *ESR1* mutations as a potential monitoring biomarker for patients with breast cancer treated with endocrine therapy. When using these molecular assays, it is important to have a comprehensive understanding of the associated principles, advantages, disadvantages, and precautions for data interpretation (Table 2).

## Figures and Tables

**Figure 1 ijms-21-08807-f001:**
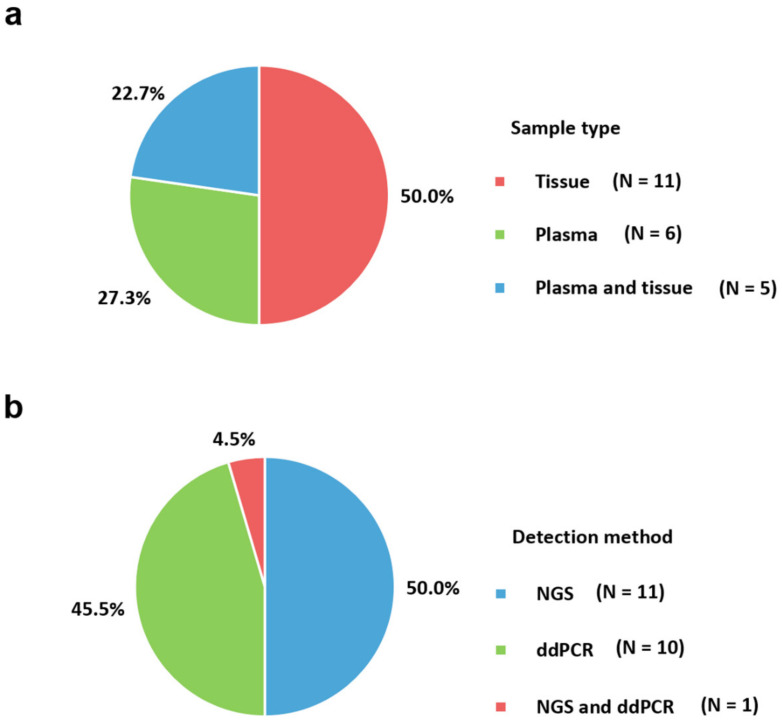
Applied laboratory assays for identifying *ESR1* mutations. (**a**) Pie charts showing the sample types used; (**b**) Pie charts showing the use of molecular detection methods. NGS, next-generation sequencing; ddPCR, droplet digital polymerase chain reaction.

**Figure 2 ijms-21-08807-f002:**
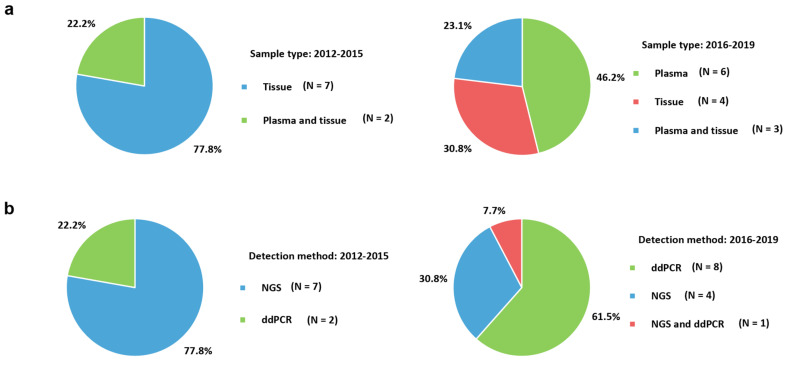
Changes in the application of molecular assays for detecting *ESR1* mutations over time. (**a**) Pie charts showing the changes in the used sample types; (**b**) Pie charts showing the changes in the use of molecular detection assays. NGS, next-generation sequencing; ddPCR, droplet digital polymerase chain reaction.

**Table 2 ijms-21-08807-t002:** Summary of commonly used molecular assays for the detection of *ESR1* mutations.

Detection Method	Principle	Advantage	Limitation
NGS platforms	Massive parallel sequencing (mostly second generation)	Higher throughput and faster time than Sanger sequencing Detecting more dynamic range of genetic alterations than ddPCR	Time-consuming for data analysis Necessity of special knowledge for bioinformatics
Illumina	Amplification: Bridge PCR Sequencing: Synthesis using fluorescently labelled reversible terminator Detection: bases from individual images using camera	Higher throughput and accuracy than Ion Torrent because of the addition of a single base at a time reducing the homopolymer sequencing error	Partial overlap between emission spectra of the fluorophores and losing activity of fluorescent dyes limiting the base calling
Ion Torrent	Amplification: Emulsion PCR Sequencing: Semi-conductor sequencing utilizing hydrogen ion detection Detection: signals using ion sensor on the complementary metal-oxide-semiconductor chip	Longer reads and easier preparation and more direct, rapid, and less expensive than Illumina relying on laser scanners	Relatively low output and higher raw error rate due to vulnerability to insertions/deletions (indels) errors associated with homopolymeric stretches and repeats when compared to Illumina
ddPCR	Partitioning the reaction through water-oil emulsion droplets (emulsion-based digital PCR) or a chip with micro-channels (microfluidics-based digital PCR) Identified of target sequences using fluorescence and subsequent calculation the initial copy number and concentration of target molecules	Application to the detection of circulating tumor DNA based on high sensitivity with a detection limitation of 0.001% Absolute quantification via micro-partitioning Increase of the tolerance of PCR system to inhibitors because of the numerous micro-compartments Highly readable two-dimensional data Reliable results with small amount of samples because of the elimination error from pre-amplification High repeatability results based on its independence on amplification efficiency Lower cost compared to NGS platforms	Narrow dynamic range for genetic alterations compared to NGS technology Relatively higher cost than real-time quantitative PCR

NGS, next-generation sequencing; ddPCR, droplet digital polymerase chain reaction.

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
