# Peer review of "Currently Applied Molecular Assays for Identifying ESR1 Mutations in Patients with Advanced Breast Cancer"

_ijms, 2020, doi:10.3390/ijms21228807_

Round 1

Reviewer 1 Report

  1. If possible include the global breast cancer statistics of 2019 or 2020 rather than 2018 in the introduction section.
  2. “The changes in nutrition, anthropometry, age at menstruation, hormone intake, lactation, and reproductive patterns, including fewer children and later age at first birth, contribute to the epidemiology of breast cancer”. Reference of A cancer statistcs 2019 by Siegel et. al is cited here which does not appear to be correct.
  3. Scientific English corrections are needed.
  4. Throughout the paper, references need a check. References are not cited at the right position.

Eg. “Aromatase inhibitors, selective estrogen receptor modulators, and selective estrogen receptor degraders are major endocrine strategies for the treatment of ER-positive advanced breast cancers.” Needs a reference. If “16” is the reference for this line, then place it after this line rather than towards the end of the paragraph.

  1. In Table 1-are there no recent studies (2019/2020) with molecular assays for ESR1 mutations?
  2. In figure 1 and 2 please include “N” or number of studies from which the piechart has been derived and not just the percentages.
  3. Bring heading “Sample Type” to position 3 and “Trends in Molecular assay” to 4.
  4. Headings such as Next generation Sequencing, Droplet Digital PCR and Other Methods should come under one single heading of “Molecular Assays” rather than independent headings.
  5. A separate table for principle, advantages and limitations of atleast Next generation Sequencing and Droplet Digital PCR should be prepared. If it can be extended to other techniques as well then it will significantly improve the paper.
  6. Conclusion and Future perspective paragraph is too long. Please break this section into Future of Molecular assays and conclusionseperately.

Reviewer 2 Report

Lee et al reviewed literature about ESR1 mutations and the molecular assays to use them for mutations detection in the clinic. Overall, the authors did a great job with the assays part. On th eother hand, earlier parts related to ESR1 alterations are not up to date. There are several studies about mutations, gene fusions, and how they can be targeted in the clinic. Please udate your references with latest articles related to these.

Author Response

Comments from the reviewers:

-Reviewer 2

Lee et al reviewed literature about ESR1 mutations and the molecular assays to use them for mutations detection in the clinic. Overall, the authors did a great job with the assays part. On the other hand, earlier parts related to ESR1 alterations are not up to date. There are several studies about mutations, gene fusions, and how they can be targeted in the clinic. Please udate your references with latest articles related to these.

  • As indicated by the reviewer, we have added recent studies for ESR1 mutations and gene fusions related to clinics to the revised 3.2.2. ESR1 Rearrangements and Fusion and 1.3.2.3. ESR1 Point Mutation as follows.

1.3.2.2. ESR1 Rearrangements and Fusion (page 4, lines 143 to 151)

“A recent study [49] showed a novel mechanism that ESR1-CCDC170 bound to HER2/HER3/SRC and activated SRC/PI3K/AKT signaling. Therefore, treatment regimens combining endocrine agents with the HER2 inhibitor lapatinib and/or the SRC inhibitor dasatinib might be applied to patients with ESR1-CCDC170 gene fusions. Furthermore, kinase fusions in breast cancer analyzed by Memorial Sloan Kettering-Integrated Mutation Profiling of Actionable Cancer Targets seemed to be enriched in hormone-resistant, metastatic carcinomas and mutually exclusive with ESR1 mutations [50]. Based on these results, fusion testing as a molecular testing at progression after endocrine therapy was suggested in an effort to identify additional therapeutic options which may provide substantial clinical benefit.”

1.3.2.3. ESR1 Point Mutation (page 5, lines 187 to 191)

“According to a recent study [62], the emergence of circulating ESR1 mutations was related to the risk of early progression during aromatase inhibitor treatment in patients with metastatic breast cancer, which suggested the potential role of ESR1 mutations as a useful biomarker. In addition, the immunogenicity of epitope derived from the most common ESR1 mutations including D538G and Y537S was suggested as novel targets for breast cancer immunotherapy [63].”

Reviewer 3 Report

The manuscript by Lee and colleagues, offer a summary of the current knowledge of the molecular assays for ESR1 mutations, one of the main cause of advanced estrogen-positive breast cancer with resistance to endocrine therapy. They investigated the advances of technical management considering time progression and literature to gain insight into the molecular methodologies applied to this challenging biological context and to facilitate the adoption of appropriate laboratory settings for ESR1 mutations evaluations in order to better predict patient’s outcomes and personalized therapeutic strategies. 

This review is well organized and presents a quite comprehensive point of view of the literature in the analysed field, however, the manuscript has some minor points that need to be addressed and clarified before it can be then considered for publication.

Minor points

- I would suggest to the authors to include more words concerning breast cancer molecular characteristics and heterogeneity as well as endocrine therapy treatments and resistance, especially with the focus on the impact of ESR1 mutations in the “Introduction section” avoiding section 2. This will give a better and more organic view of the ESR1 mutations in the main breast cancer biological contexts such as therapeutic strategy and resistance also to the readers not familiar with this field.

- Concerning Table 1, I would suggest to the authors to modify it a bit for a better understanding. I would include a sub classification according methodologies applied to investigate/detect ESR1 mutations.

- I would encourage the authors to resize (increase) labels (detection methods and simple size) in the Figures (1 and 2) for a better reading.

There are some inaccuracies and typos within the text that need to be corrected.

Round 2

Reviewer 1 Report

The auhors have addressed my comments well. The manuscript can be published.

Reviewer 2 Report

They answered my comments.